# Evolutionary adaptation of an HP1-protein chromodomain integrates chromatin and DNA sequence signals

**Lisa Baumgartner[1,2], Jonathan J Ipsaro[3], Ulrich Hohmann[1,4], Dominik Handler[1], Alexander Schleiffer[1,4], Peter Duchek[1], Julius Brennecke[1]***

[1]Institute of Molecular Biotechnology of the Austrian Academy of Sciences (IMBA), Vienna BioCenter (VBC), Vienna, Austria; [2]Vienna BioCenter PhD Program, Doctoral School of the University of Vienna and Medical University of Vienna, Vienna, Austria; [3]Howard Hughes Medical Institute, W.M. Keck Structural Biology Laboratory, Cold Spring Harbor Laboratory, Cold Spring Harbor, New York, United States; [4]Research Institute of Molecular Pathology (IMP), Vienna BioCenter (VBC), Vienna, Austria

**\*For correspondence:**
julius.brennecke@imba.oeaw.ac.at

**Competing interest:** The authors declare that no competing interests exist.

**Abstract** Members of the diverse heterochromatin protein 1 (HP1) family play crucial roles in heterochromatin formation and maintenance. Despite the similar affinities of their chromodomains for di- and tri-methylated histone H3 lysine 9 (H3K9me2/3), different HP1 proteins exhibit distinct chromatin-binding patterns, likely due to interactions with various specificity factors. Previously, we showed that the chromatin-binding pattern of the HP1 protein Rhino, a crucial factor of the *Drosophila* PIWI-interacting RNA (piRNA) pathway, is largely defined by a DNA sequence-specific $C_2H_2$ zinc finger protein named Kipferl (Baumgartner et al., 2022). Here, we elucidate the molecular basis of the interaction between Rhino and its guidance factor Kipferl. Through phylogenetic analyses, structure prediction, and in vivo genetics, we identify a single amino acid change within Rhino's chromodomain, G31D, that does not affect H3K9me2/3 binding but disrupts the interaction between Rhino and Kipferl. Flies carrying the *rhino*[G31D] mutation phenocopy *kipferl* mutant flies, with Rhino redistributing from piRNA clusters to satellite repeats, causing pronounced changes in the ovarian piRNA profile of *rhino*[G31D] flies. Thus, Rhino's chromodomain functions as a dual-specificity module, facilitating interactions with both a histone mark and a DNA-binding protein.

## eLife assessment

This **fundamental** work has completed our understanding of the singular binding profile of the Rhino HP1 protein to chromatin, a key step in converting certain genomic regions into piRNA source loci. The evidence supporting the conclusions is **compelling**. Phylogenetic analyses, structure prediction, rigorous biochemical assays and in vivo genetics emphasize the importance of the Rhino chromodomain in the recognition of both a histone mark and a DNA-binding protein, and highlight the importance of a single chromodomain residue in the protein-protein interaction.

## Introduction

Transposable elements (TEs) are ubiquitous in eukaryotic genomes but pose a significant threat to genome integrity (*Bourque et al., 2018*). When activated and mobile, these selfish genetic elements can lead to insertional mutagenesis and ectopic recombination events, imposing significant fitness costs on their hosts. To counteract the deleterious effects of TEs, eukaryotes package TE loci into repressive heterochromatin, effectively silencing these elements and preventing their uncontrolled

movement within the genome (*Levin and Moran, 2011*; *Fedoroff, 2012*). Proteins of the heterochromatin protein 1 (HP1) family play a central role in the initiation and maintenance of heterochromatin from fungi to animals (*Vermaak and Malik, 2009*).

The founding member of the HP1 family, *Drosophila* Su(var)2–5, acts as a strong suppressor of position effect variegation (*James and Elgin, 1986*; *Eissenberg et al., 1990*; *Eissenberg et al., 1992*). It binds to heterochromatic histone marks and facilitates transcriptional silencing and the compaction of chromatin through the recruitment of histone methyltransferases, histone deacetylases, and other repressive activities (*Vermaak and Malik, 2009*; *Allshire and Madhani, 2018*). Most animal genomes encode multiple HP1 homologs that share a common domain architecture. They contain an N-terminal chromodomain with specific affinity for di- and tri-methylated histone H3 lysine 9 (H3K9) peptides (*Bannister et al., 2001*; *Lachner et al., 2001*), an unstructured central hinge region of variable length involved in nonspecific nucleic acid interactions (*Keller et al., 2012*), and a C-terminal chromo shadow domain (*Aasland and Stewart, 1995*). While resembling the chromodomain fold, the chromo shadow domain does not bind histone tails. Instead, it forms a dimerization interface with the chromo shadow domain of another HP1 protein, creating a binding groove for proteins containing a PxV/LxL consensus motif (*Smothers and Henikoff, 2000*).

The number of HP1 family members varies between species. For instance, mice and humans encode three HP1 family proteins (HP1α, HP1β, and HP1γ), whereas *Drosophila melanogaster* encodes five different members: the ubiquitously expressed HP1a/Su(var)2–5, HP1b, and HP1c proteins, and the germline-specific HP1d/Rhino (ovary and testis) and HP1e (testis) proteins (*Vermaak and Malik, 2009*; *Levine et al., 2012*). Despite having similar affinities for H3K9me2/3 reported from in vitro experiments, the *Drosophila* HP1 proteins have distinct biological functions and chromatin-binding patterns (*Yu et al., 2015*; *Lee et al., 2019*; *Baumgartner et al., 2022*). For example, while Su(var)2–5 binds all H3K9-methylated loci genome-wide, the germline-specific Rhino is enriched only at specific heterochromatic loci from where the non-coding precursors of PIWI-interacting RNAs (piRNAs) are transcribed (*Vermaak et al., 2005*; *Klattenhoff et al., 2009*; *Mohn et al., 2014*; *Zhang et al., 2014*). These so-called piRNA clusters are rich in repetitive sequences and serve as heritable sequence storage units that confer specificity to the piRNA pathway, a small RNA-based TE silencing system in animal gonads (*Brennecke et al., 2007*; *Czech et al., 2018*; *Ozata et al., 2019*). At the molecular level, Rhino facilitates the productive expression of heterochromatic piRNA clusters by recruiting specific effector proteins that stimulate transcription initiation, elongation, and nuclear export of the resulting non-coding piRNA precursors (*Klattenhoff et al., 2009*; *Mohn et al., 2014*; *Zhang et al., 2014*; *Chen et al., 2016*; *Andersen et al., 2017*; *ElMaghraby et al., 2019*; *Kneuss et al., 2019*). This makes Rhino a remarkably specialized HP1 protein that mediates activating, rather than repressive, chromatin identity. The precise regulation of Rhino's chromatin-binding profile is therefore of great importance.

The zinc finger protein Kipferl, one of about 90 ZAD zinc finger proteins in *Drosophila*, acts as a critical guidance factor for Rhino in ovaries (*Baumgartner et al., 2022*). Kipferl binds to chromatin at genomic sites enriched in GRGGN motifs, presumably through a direct interaction between its $C_2H_2$ zinc finger arrays and DNA. When genomic Kipferl-binding sites are located within an H3K9me2/3 domain, Kipferl recruits Rhino, and both proteins form extended binding domains around initial nucleation sites. The interaction between Kipferl and Rhino occurs between Kipferl's fourth zinc finger and Rhino's chromodomain. This interaction represents a highly unusual mode of binding because, unlike other interactions with HP1 proteins, it does not involve the dimeric HP1 chromo shadow domain.

Here, we reveal the molecular basis underlying the interaction between Kipferl and the Rhino chromodomain. We identified a single amino acid adaptation within Rhino's chromodomain that discriminates it from other HP1 family members and is critical for the specific Kipferl–Rhino interaction. Our findings provide important insights into how a direct protein–protein interaction dictates the chromatin-binding profile of an HP1 protein, demonstrating how a single amino acid residue can contribute to the emergence of a novel protein function.

## Results and discussion

### Phylogenetic and structure prediction analyses of the Rhino–Kipferl interaction

Previous yeast two-hybrid (Y2H) experiments demonstrated a direct interaction between Kipferl and the chromodomain of Rhino, but not with that of the related Su(var)2–5 protein (*Figure 1A*; *Baumgartner et al., 2022*). To explore the binding specificity of Kipferl for Rhino, we conducted a comparative phylogenetic analysis of the chromodomains of Rhino, Su(var)2–5, HP1b, HP1c and HP1e homologs from various *Drosophila* species with clearly identified Kipferl orthologs (*Figure 1B*; *Figure 1—figure supplement 1*, *Figure 1—figure supplement 2*). This analysis highlighted two Rhino-specific and conserved sequence alterations: the D31G change and the G62 insertion (*Figure 1B*).

To explore whether either of the two Rhino-specific residues might contribute to the interaction with Kipferl, we used AlphaFold2 Multimer (*Jumper et al., 2021*; *Evans et al., 2022*) to predict interactions between Rhino's chromodomain and Kipferl's first zinc finger array, which comprises four $C_2H_2$ zinc fingers and was identified as the interaction site with Rhino (*Baumgartner et al., 2022*). Alpha-Fold2 Multimer predicted a high confidence interaction with a single conformation in 5/5 models, involving the fourth zinc finger of Kipferl, which is necessary and sufficient for the Y2H interaction with Rhino (*Figure 1C*, *Figure 1—figure supplement 3A–C*; *Baumgartner et al., 2022*). No interaction was predicted between Kipferl and the chromodomains of Su(var)2–5, HP1b, HP1c, or HP1e. The predicted Kipferl–Rhino complex is compatible with binding to the H3K9me2/3 peptide through Rhino's aromatic cage (*Figure 1C*) and would allow for a potential dimerization of the Rhino chromodomain (*Yu et al., 2015*).

In the predicted complex, Kipferl's fourth zinc finger interacts with Rhino's chromodomain through an extended interface opposite the aromatic cage, including β-sheets 2–4 and the C-terminal α-helix of Rhino's chromodomain (*Figure 1C*, *Figure 1—figure supplements 1 and 4*). While the Rhino-specific G62 insertion does not participate in contacts with Kipferl, the Rhino-specific G31 residue, which in other HP1 proteins is a highly conserved aspartic acid, is centrally located in the predicted interaction interface (*Figure 1C*, *Figure 1—figure supplement 1*). Due to the nature of the predicted Kipferl–Rhino interaction, substituting glycine with aspartic acid at position 31 in Rhino would cause steric clashes with Kipferl residues V285 and F286, preventing the association of both proteins. We therefore hypothesized that mutating Rhino G31 to the HP1-typical aspartic acid residue (Rhino[G31D]) should specifically uncouple Rhino and Kipferl while leaving Rhino otherwise functionally intact.

### The Rhino[G31D] chromodomain retains H3K9me3-binding in vitro

Rhino's in vivo function depends critically on its ability to bind H3K9me2/3 via its chromodomain (*Yu et al., 2015*). In addition, dimerization of the Rhino chromodomain has been suggested to be important for its function (*Yu et al., 2015*). To determine whether the Rhino G31D mutation affects either of these functions, we analyzed a panel of recombinantly expressed Rhino chromodomains. This panel included the wildtype construct, two putative Kipferl-binding mutants (G31A and G31D), and control mutants that impair H3K9me2/3 binding (mutations of the aromatic cage residues Y24A, W45A, or F48A) or putative dimerization (F34A/F76A double mutant) (*Yu et al., 2015*).

We used analytical size-exclusion chromatography with inline multiangle light scattering (SEC-MALS) to assess the oligomeric state of the different Rhino chromodomain constructs. Our data confirmed differences in elution volume among the different mutant constructs (*Yu et al., 2015*), but these differences did not correspond to significant changes in their in-solution molecular weight, indicating that the oligomeric state remained consistent across all constructs tested (*Figure 2A*; *Figure 2—figure supplement 1*). We conclude that the isolated wildtype Rhino chromodomain, along with the G31D or G31A variants, are monomeric in solution, as has been shown for other HP1 homologs (*Jacobs et al., 2001*; *Brasher et al., 2000*). To further investigate whether the G31D mutation causes any unwanted structural changes in the Rhino chromodomain, we performed circular dichroism spectroscopy. All tested mutant constructs exhibited similar secondary structure compositions compared to the wild-type construct (*Figure 2—figure supplement 2*).

Having established that the two G31 mutant chromodomains do not exhibit altered protein folding or oligomeric state, we tested both constructs for their ability to bind H3K9me3 peptides alongside wildtype and aromatic cage mutant (F48A) controls. Consistent with previous observations (*Yu et al., 2015*; *Le Thomas et al., 2014*), isothermal titration calorimetry experiments using synthetic H3K9me3

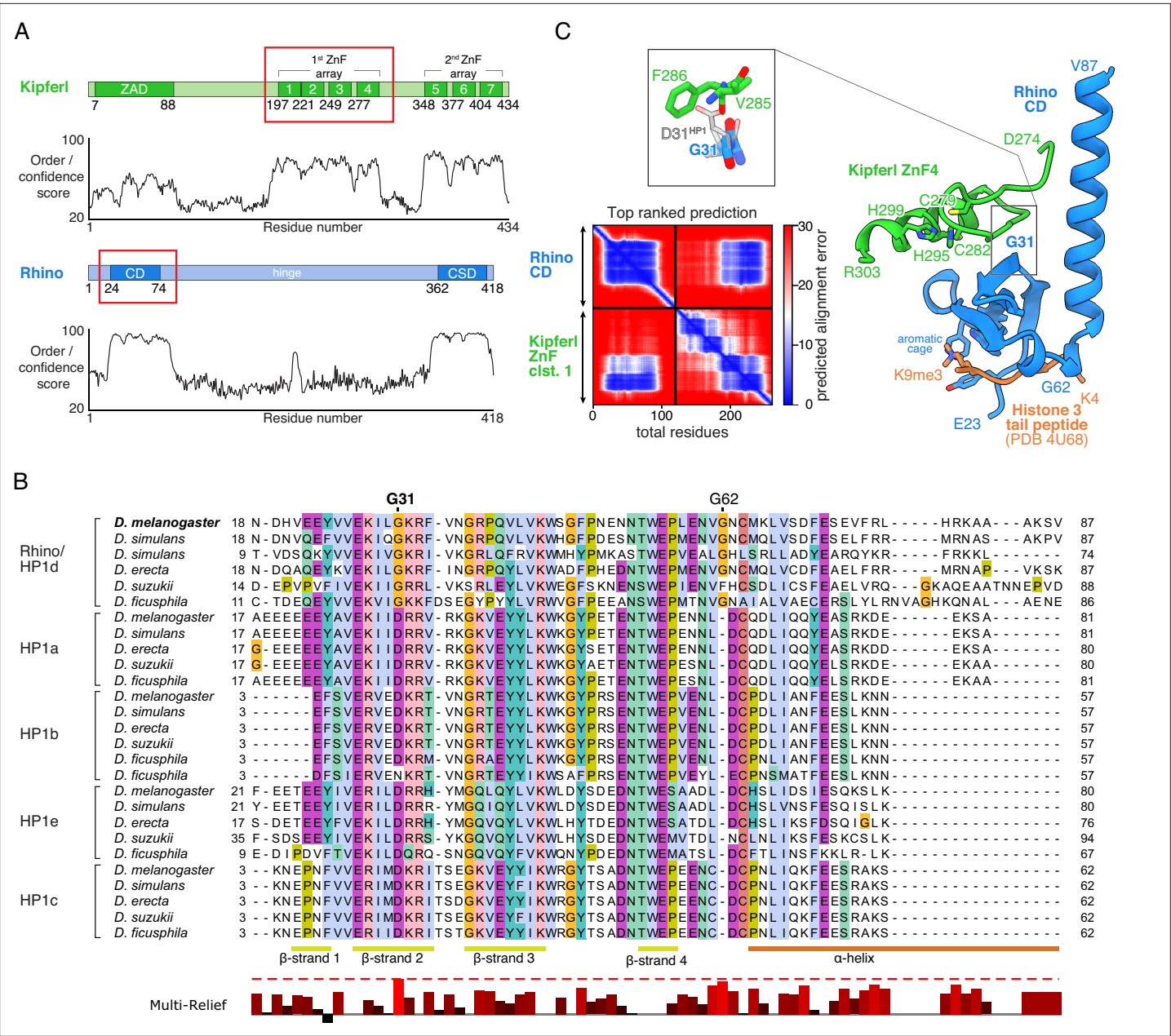

**Figure 1.** Structure prediction and phylogenetic analyses point to a Rhino-specific residue involved in binding Kipferl. (**A**) Domain organization of Kipferl and Rhino, with the AlphaFold2 Multimer predicted local distance difference test (pLDDT) score plotted as a measure of order or disorder alongside. Red boxes indicate the smallest interacting fragments identified by yeast two-hybrid experiments by **Baumgartner et al., 2022**. ZAD, zinc finger-associated domain; ZnF, zinc finger; CD, chromodomain; CSD, chromo shadow domain. (**B**) Multiple sequence alignment of heterochromatin protein 1 (HP1) family proteins in five selected species harboring an unequivocally identified Kipferl homolog (see **Figure 1—figure supplement 2**). Rhino-specific amino acid residues are indicated. Protein accessions and identifiers are documented in **Supplementary file 1**. Multi-Relief representation indicates residues that differ significantly in Rhino homologs versus other HP1 variant proteins. Note that two Rhino paralogs are identified in *D. simulans* (see **Supplementary file 1** for accessions). (**C**) Predicted aligned error (PAE) plot for the top ranked AlphaFold2 Multimer prediction of the Rhino chromodomain with the Kipferl ZnF cluster 1 (left) and structure of the complex in cartoon representation (Rhino in blue; Kipferl in green), together with the H3K9me3 peptide (orange) as observed in a Rhino–H3K9me3 crystal structure (PDB ID 4U68). Key residues of Rhino's aromatic cage and H3K9me3, as well as of Kipferl's $C_2H_2$ ZnF4 are shown in sticks representation. Only the interacting ZnF4 is shown. Depicted in the inset are Rhino G31 and HP1 D31, with HP1 (PDB ID 6MHA) superimposed on Rhino chromodomain residues 26–57 (root mean square deviation = 0.55 Å), together with Kipferl V285 and F286, illustrating that D31 would lead to steric clashes with Kipferl.

The online version of this article includes the following figure supplement(s) for figure 1:

**Figure supplement 1.** Multiple sequence alignment of heterochromatin protein 1 (HP1) family proteins across *Drosophila* species.

*Figure 1 continued on next page*

*Figure 1 continued*

**Figure supplement 2.** Phylogenetic tree illustrating the evolutionary relationship of zinc finger-associated domain (ZAD)-containing zinc finger proteins based on ZAD protein sequence.

**Figure supplement 3.** Diagnostic plots for ranks 1–5 for the AlphaFold2 Multimer prediction of the Rhino chromodomain with the Kipferl ZnF cluster 1.

**Figure supplement 4.** Multiple sequence alignment of Kipferl proteins across *Drosophila* species.

peptides revealed an affinity of 30.9 ± 3.0 µM for the wildtype domain and no measurable affinity for the F48A mutant (*Figure 2B*). Despite slight changes in the thermodynamic binding parameters, both the G31A and G31D mutants showed affinities comparable to the wildtype constructs with 43.5 ± 8.6 and 31.1 ± 3.2 µM, respectively. Thus, the Rhino^G31D chromodomain behaves similar to the wildtype domain in terms of oligomeric state, folding, and ability to bind H3K9me3 peptides in vitro.

## The *rhino^G31D* mutant uncouples Rhino from Kipferl

To explore the importance of G31 for Rhino function in vivo, we engineered a single point mutation within the endogenous *rhino* locus, converting G31 to the aspartic acid residue typically present in all other HP1 proteins (*rhino^G31D*). In *kipferl* mutant females, Rhino fails to localize to the majority of its genomic binding sites, resulting in diminished piRNA levels and compromised fertility (*Baumgartner et al., 2022*). Homozygous females carrying the *rhino^G31D* allele were viable but exhibited severely reduced fertility: Although the egg-laying rate of *rhino^G31D* females was comparable to that of control flies, the hatching rate of laid eggs dropped to 21 ± 9% (*Figure 3A*). While this decline in fertility was less severe compared to the complete sterility observed in *rhino* null mutants, it closely mirrored the impaired fertility of *kipferl* null mutants, which was in the range of 15 to 40% (*Baumgartner et al., 2022*), providing a first indication that the G31D mutation may specifically affect the Rhino–Kipferl interaction.

To gain deeper insights into the Rhino–Kipferl interaction in *rhino^G31D* mutants, we examined changes to the pronounced colocalization of Kipferl and Rhino at discrete nuclear foci – corresponding to piRNA source loci – observed in wildtype nurse cells (*Baumgartner et al., 2022*). Using

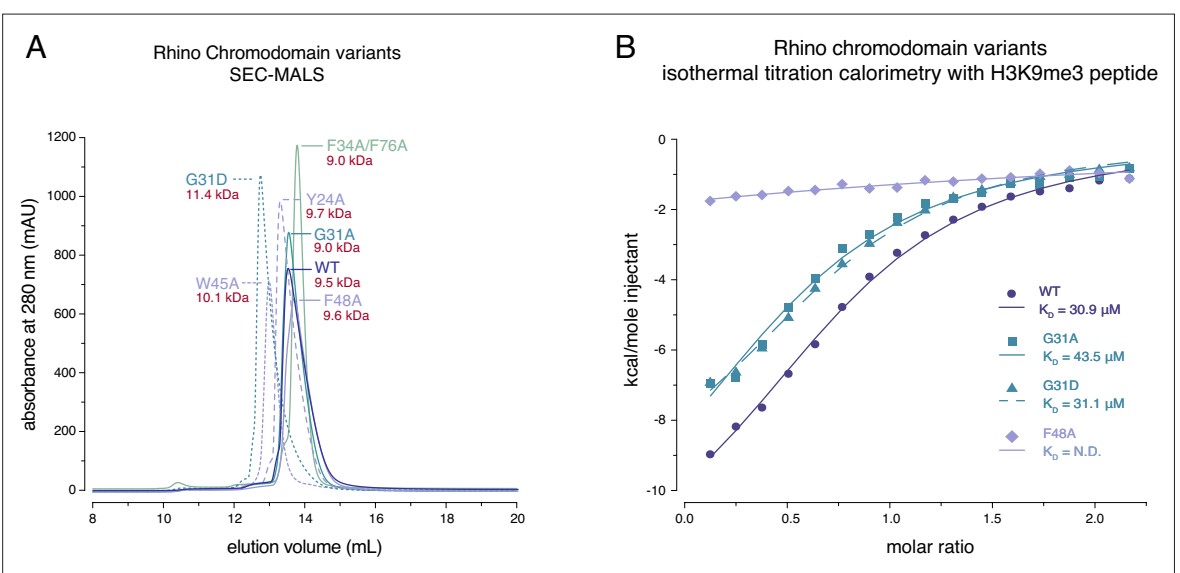

**Figure 2.** Rhino G31 point mutations do not affect Rhino's ability to bind H3K9me3. (**A**) Line graph summarizing size-exclusion chromatography with inline multiangle light scattering (SEC-MALS) results for the examined Rhino chromodomain constructs. The in solution molecular weight is indicated for each construct. (**B**) Isothermal titration calorimetry results showing the binding of indicated Rhino chromodomain constructs to the H3K9me3-modified histone tail peptide.

The online version of this article includes the following figure supplement(s) for figure 2:

**Figure supplement 1.** Individual line graphs depicting size-exclusion chromatography with inline multiangle light scattering (SEC-MALS) results for the examined Rhino chromodomain constructs with in solution molecular weight measurements depicted in red.

**Figure supplement 2.** Purified Rhino point mutant chromo domains display normal protein folding.

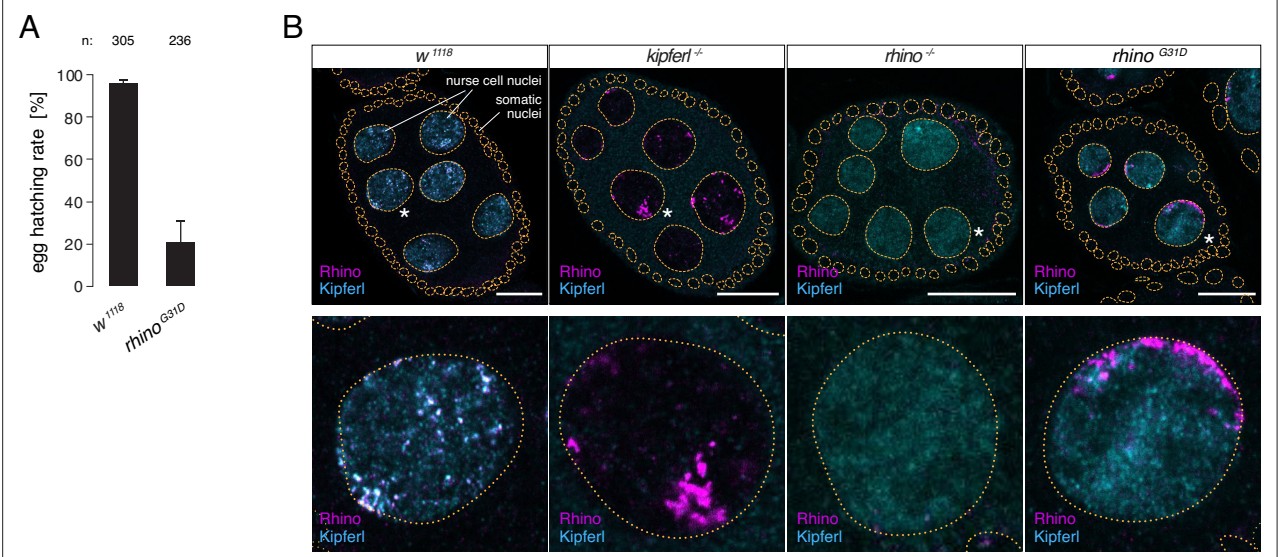

**Figure 3.** The *rhino*$^{G31D}$ point mutation recapitulates the phenotypes for Rhino and Kipferl in each other's null-mutant background. (**A**) Bar graph depicting female fertility as egg hatching rate in percent of laid eggs for indicated genotypes. Error bars indicate standard deviation of three technical replicates. The total number of eggs assessed across all replicates is given as n. (**B**) Confocal images showing immunofluorescence signal for Kipferl and Rhino in egg chambers of indicated genotypes. Zoomed images display one representative nurse cell nucleus (labeled by white asterisk in panel A) per genotype (scale bar: 20 µm).

immunofluorescence imaging, we observed a complete absence of colocalization between Rhino and Kipferl in *rhino*$^{G31D}$ mutants (**Figure 3B**). Kipferl localized diffusely in the nucleus with only a few foci, mirroring its distribution in *rhino* null mutants. Rhino$^{G31D}$ was not enriched within these Kipferl foci; instead, it accumulated in prominent structures near the nuclear envelope, resembling the Rhino accumulations found in *kipferl* null mutants (**Baumgartner et al., 2022**).

To determine the chromatin-binding patterns of Rhino and Kipferl in ovaries of *rhino*$^{G31D}$ mutant flies, we performed chromatin immunoprecipitation followed by sequencing (ChIP-seq). In wildtype ovaries, Rhino and Kipferl co-occupy hundreds of heterochromatic domains, displaying nearly identical enrichment patterns (**Figure 4A**; **Baumgartner et al., 2022**). In addition, Kipferl, but not Rhino, binds to specific sites in euchromatin (Kipferl-only sites) that lack H3K9me2/3 marks but are enriched in GRGGN motifs, Kipferl's presumed DNA-binding motif. To account for the heterogeneous size of genomic Rhino/Kipferl domains, we analyzed their binding profiles by quantifying genome-unique ChIP-seq reads mapped to non-overlapping genomic 1-kb tiles (**Mohn et al., 2014**). In *kipferl* mutants, Rhino is lost from most of its genomic binding sites, with retained Rhino binding primarily corresponding to piRNA clusters *38C* and *42AB* (**Figure 4A, B**; **Baumgartner et al., 2022**). Conversely, in *rhino* mutants, Kipferl binding persists at euchromatic Kipferl-only sites but is strongly reduced at loci that are co-occupied by Kipferl and Rhino in wildtype: at sites where Rhino binding is Kipferl dependent, Kipferl binding is reduced to more defined, narrow peaks. At Kipferl-independent loci on the other hand (e.g., piRNA clusters *38C* and *42AB*), Kipferl binding is almost completely lost in *rhino* mutants (**Figure 4A**). ChIP-seq experiments in *rhino*$^{G31D}$ mutant ovaries revealed a chromatin occupancy for Rhino$^{G31D}$ that was almost indistinguishable from that of wildtype Rhino in *kipferl* mutants (**Figure 4C, D**). This similarity extended to Kipferl-independent loci (e.g., piRNA clusters *38C* and *42AB*), where the altered chromatin occupancy of Rhino in *kipferl* mutants was mirrored by Rhino$^{G31D}$ (**Figure 4A**). At the same time, the chromatin-binding profile of Kipferl in *rhino*$^{G31D}$ mutants strongly resembled that observed in *rhino* null-mutants genome wide (**Figure 4A, E**). Taken together, the mutation of a single Rhino-specific chromodomain residue to its ancestral state results in the functional uncoupling of Rhino and Kipferl at the molecular level.

## Rhino$^{G31D}$ is functional at Kipferl-independent piRNA source loci

To assess the impact of the Rhino$^{G31D}$ point mutation on Rhino's overall functionality, we analyzed Kipferl-independent but Rhino-dependent piRNA source loci. In *kipferl* mutant ovaries, Rhino is

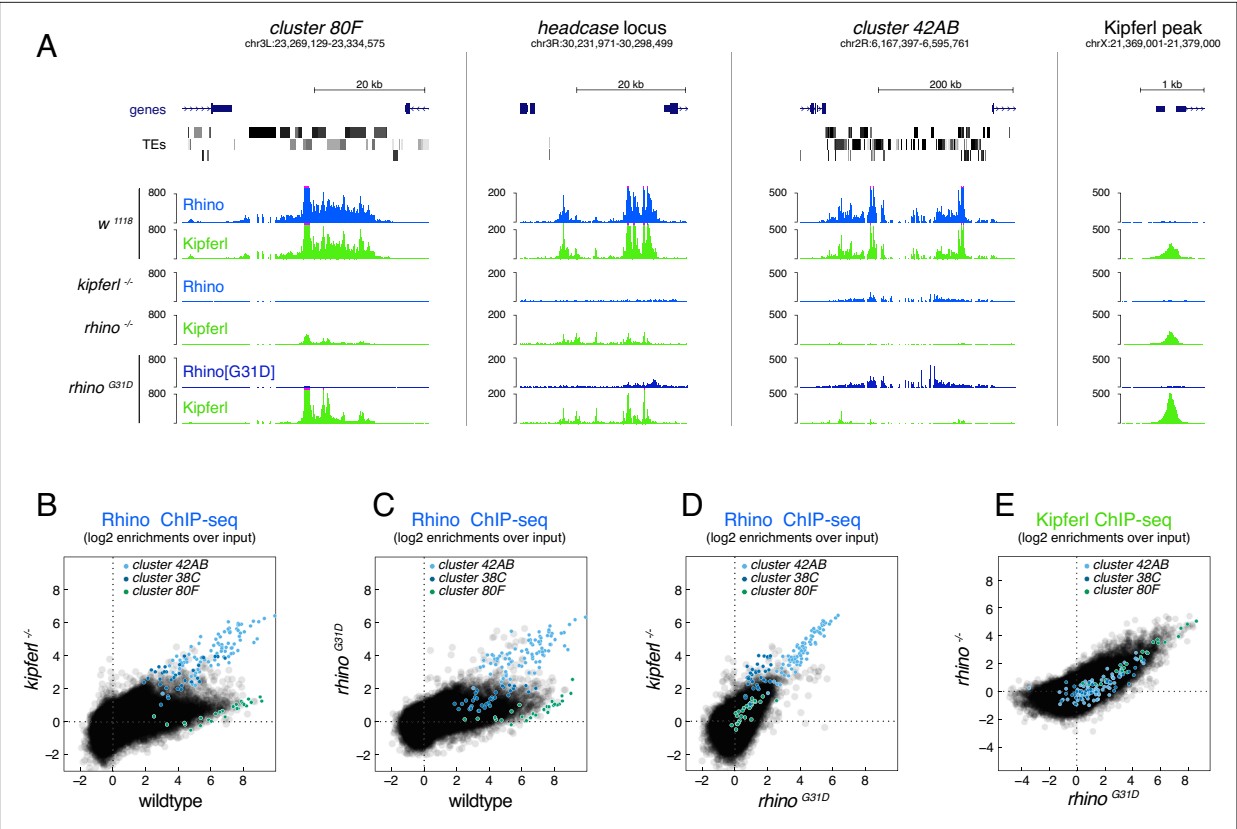

**Figure 4.** The Rhino[G31D] point mutation uncouples Rhino and Kipferl on chromatin. (**A**) UCSC genome browser screenshots depicting the ChIP-seq signal for Rhino and Kipferl at diverse Rhino domains in ovaries of the indicated genotypes (signal shown as coverage per million sequenced reads for one representative replicate). Scatter plot of genomic 1-kb tiles contrasting average log2-fold ChIP-seq enrichment for Rhino (**B–D**) or Kipferl (**E**) in ovaries of the indicated genotypes (values displayed represent the average of two to three replicate experiments).

sequestered to large DNA satellite arrays, resulting in greatly increased transcription and piRNA production at the *Responder* and *1.688* g/cm³ family satellites (*Baumgartner et al., 2022*). In *rhino[G31D]* mutants, RNA fluorescent in situ hybridization (FISH) experiments showed that transcription of the *Rsp* and *1.688* g/cm³ satellites was also strongly elevated, leading to elongated structures at the nuclear envelope, reminiscent of the phenotype observed in *kipferl* mutant nurse cell nuclei (*Figure 5A*). Consistent with this elevated transcription, Rhino[G31D] was enriched at satellite consensus sequences as determined by ChIP-seq, while it was reduced at most transposon sequences (*Figure 5B, C*). These findings extended to piRNA levels: piRNAs originating from *Rsp* and *1.688* g/cm³ satellites were substantially increased (*Figure 5D*), while piRNAs were reduced at Kipferl-dependent piRNA clusters (e.g., *cluster 80F*), but not at Kipferl-independent piRNA clusters like *38C* and *42AB* (*Figure 5E*). Similarly, the levels of piRNAs mapping to transposon consensus sequences showed similar behaviors in *rhino[G31D]* mutants as observed in *kipferl* mutants (*Figure 5—figure supplement 1A*). This provides further confirmation that the Rhino[G31D] mutation faithfully phenocopies a *kipferl* null-mutant, indicating that Rhino[G31D] remains fully functional at Kipferl-independent loci. The altered piRNA levels observed in *kipferl* mutant ovaries result in the de-repression of a handful of TEs (*Baumgartner et al., 2022*). Based on RNA FISH experiments, the same transposons were also de-repressed in *rhino[G31D]* females, with the levels of upregulation resembling those in *kipferl* mutants rather than *rhino* mutants (*Figure 5—figure supplement 2*), further suggesting a specific requirement of G31 for Kipferl-dependent functions of Rhino.

Rhino also plays a role in specifying piRNA source loci in the male germline, where Kipferl is not expressed (*Chen et al., 2021*; *Chen and Aravin, 2023*; *Baumgartner et al., 2022*). The piRNA source loci in testes only partially overlap with those of ovaries and are dynamically regulated during spermatogenesis, suggesting Kipferl-independent mechanisms for Rhino recruitment to chromatin. To assess a potential impact of the G31D mutation on Rhino function in males, we sequenced testes small

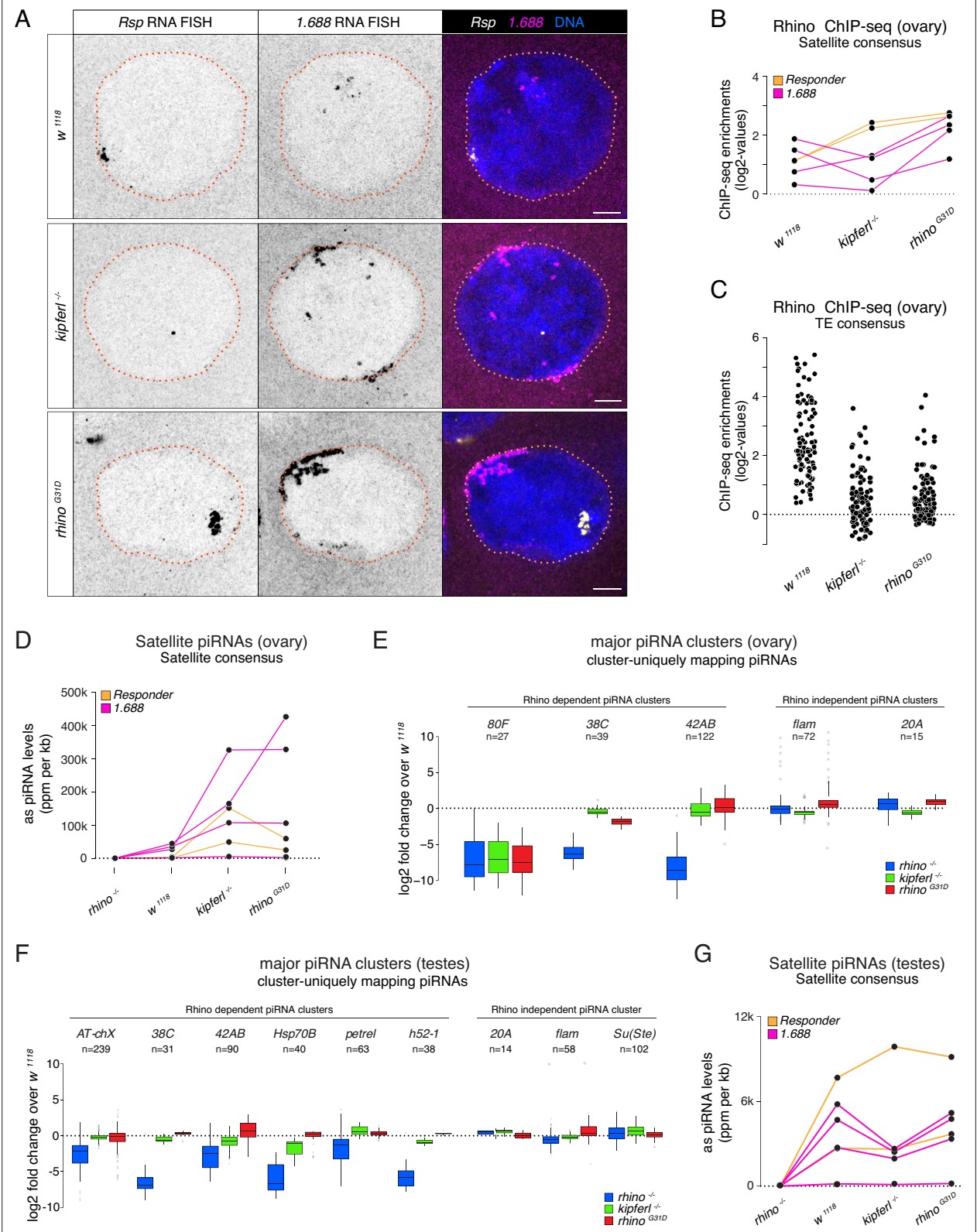

**Figure 5.** Kipferl-independent functions of Rhino are not affected by the G31D mutation. (**A**) Confocal images showing *Rsp* and *1.688g/cm³* Satellite RNA fluorescent in situ hybridization (FISH) signal in nurse cells of indicated genotypes (scale bar: 5 μm). Jitter plots depicting the log2-fold enrichments for Rhino ChIP-seq on consensus sequences of Satellites (**B**) or Rhino-dependent transposons (**C**) in ovaries with indicated genetic backgrounds. Jitter plots depicting the length-normalized antisense PIWI-interacting RNA (piRNA) counts on Satellite consensus sequences derived from ovaries (**D**) or

*Figure 5 continued on next page*

*Figure 5 continued*

testes (**G**) of indicated genetic backgrounds. Box plots depicting the log2-fold change of piRNA counts (compared to $w^{1118}$ control) per 1-kb tile for major piRNA clusters in ovaries (**E**) or testes (**F**) of the indicated genotypes. The number of tiles per piRNA cluster is indicated (*n*).

The online version of this article includes the following figure supplement(s) for figure 5:

**Figure supplement 1.** Antisense piRNA levels in rhino$^{G31D}$ mutants resemble those in kipferl null mutant.

**Figure supplement 2.** Transposon upregulation visualized by RNA fluorescent in situ hybridization (FISH).

RNAs from a panel of different genetic mutants. Comparing piRNAs from *rhino* mutant testes to wild-type controls confirmed the expected loss of piRNA production specifically from dual-strand piRNA source loci, whereas piRNA levels from the same loci remained unchanged in *kipferl* or *rhino*$^{G31D}$ mutants (***Figure 5F***). Consistent with this, the levels of transposon-mapping piRNAs also remained unaltered in *kipferl* or *rhino*$^{G31D}$ mutants (***Figure 5—figure supplement 1B***). The lack of a piRNA phenotype in testes further extended to the *Rsp* and *1.688* g/cm$^3$ satellite loci, which produce Rhino-dependent piRNAs also in the male germline (***Figure 5G***). Taken together, the G31D mutation, while completely uncoupling Rhino from Kipferl, does not impede Rhino function at Kipferl-independent sites in either ovaries or testes.

## Conclusion

In this study, we elucidate the intricate interplay between the DNA sequence-specific zinc finger protein Kipferl, and the chromodomain of the HP1 variant Rhino. Our findings underscore the critical role of Kipferl in orchestrating Rhino's chromatin-binding dynamics and subsequent piRNA production. Specifically, we show that a single amino acid alteration within Rhino's chromodomain, reverting it to its ancestral state (G31D), disrupts Kipferl's ability to target Rhino to chromatin. Notably, the G31 residue in Rhino is highly conserved among *Drosophilids*, even in species that lack a clearly identifiable Kipferl ortholog. This may indicate that other proteins use a mechanism similar to Kipferl to define Rhino's chromatin occupancy in more distantly related *Drosophila* species. Our data also show that the Rhino$^{G31D}$ mutation does not affect the chromatin binding or the function of Rhino at Kipferl-independent piRNA source loci in ovaries and testes, suggesting the existence of other, G31-independent mechanisms for recruitment of Rhino to chromatin. Whether these alternative mechanisms act in a similar way to the one described here, utilizing zinc finger proteins and interactions with the Rhino chromodomain, remains an open question. An important issue for future investigation, currently hampered by the challenges of obtaining soluble recombinant Kipferl protein, will be to determine the precise three-dimensional arrangement of the Kipferl–Rhino complex together with Kipferl motif-containing DNA and H3K9-methylated nucleosomes, considering that Kipferl and Rhino are both likely to form homodimers via their N-terminal ZAD domain and C-terminal chromo shadow domain, respectively.

## Materials and methods

**Key resources table**

| Reagent type (species) or resource | Designation | Source or reference | Identifiers | Additional information |
|---|---|---|---|---|
| Antibody | anti-CG2678#2 (Rabbit polyclonal) | *Baumgartner et al., 2022* | CG2678#2_4P39glyc, raised against Kipferl peptide R171-I190 | Anti-Kipferl polyclonal antibody, available from Brennecke lab; ChIP (7 µl per IP) |
| Antibody | anti-CG2678 M3 (Mouse monoclonal) | *Baumgartner et al., 2022* | M3 2C5-3C3, raised against Kipferl amino acids M2-K188 | Anti-Kipferl monoclonal IF antibody, available from Brennecke lab; IF (1:500) |
| Antibody | anti-Rhino (Rabbit polyclonal) | *Mohn et al., 2014* | Rhino#1_3573gly | ChIP (5 µl per IP), IF (1:1000) |
| Genetic reagent (*D. melanogaster*) | w1118;;; | Bloomington stock 3605 | w1118 | Wildtype, cultivated in our lab for several years |

*Continued on next page*

*Continued*

| Reagent type (species) or resource | Designation | Source or reference | Identifiers | Additional information |
|---|---|---|---|---|
| Genetic reagent (*D. melanogaster*) | w;; CG2678[Δ1](dsRed+)/TM3,Sb; | *Baumgartner et al., 2022* | Kipferl (CG2678) | Kipferl mutant allele, available from VDRC; LB1-RMCEm31 |
| Genetic reagent (*D. melanogaster*) | w;; CG2678[fs1]/TM3,Sb; | *Baumgartner et al., 2022* | Kipferl (CG2678) | Kipferl mutant allele, available from VDRC; LB1-FSm52; indel (–7); sequence CCTGCGTCCTGGCCGTGC-------TTTCCGGTTCAAGTGGCAAAGCGAGCAGAG |
| Genetic reagent (*D. melanogaster*) | w; rhi[18-7]/CyO;; | *Andersen et al., 2017*, VDRC-ID 313488 | Rhino (CG10683) | Mutant allele |
| Genetic reagent (*D. melanogaster*) | w; rhi[g2m11]/CyO;; | *Baumgartner et al., 2022* | Rhino (CG10683) | Rhino mutant allele, available from VDRC; indel –7; seq: ATGTCTCGCAACCA-------cc-AATCTTGGTCTGGTCG ATGCACCGCCTAATG |
| Genetic reagent (*D. melanogaster*) | w; rhi[G31D]/CyO;; | This paper | Rhino (CG10683) | Mutant allele changing glycine at position 31 to aspartic acid; m3-5 |

## Fly strains and husbandry

All fly stocks were maintained at 25°C with 12 hr dark/light cycles. Fly strains used in this study are listed in the Key Resource Table. For ovary dissections, flies were aged for 2–6 days and held in cages with apple juice plates and fresh yeast paste for 2 days. Flies harboring the *rhino*[G31D] point mutation were generated from isogenized $w^{1118}$ embryos by co-injecting the pDCC6b plasmid (*Gokcezade et al., 2014*) expressing a gRNA (TATGTAGTGGAGAAAATCTT) with an HDR donor oligo (GGTC GATGCACCGCCTAAtGATCATGTCGAAGAATATGTAGTGGAGAAAATCcTgGatAAACGGTTTGTTAA TGGGCGTCCCCAGGTTCTGGTGAAGTGGAGCGGTTTTCCG; IDT).

## Phylogenetic analyses

Kipferl and related zinc finger-associated domain-containing (ZAD) proteins were collected with NCBI BLAST searches using *D. melanogaster* Kipferl ZAD (regions 5–95) in the NCBI non-redundant protein or the UniProt reference proteomes databases (*Altschul et al., 1997*; *Uniprot, 2021*; *Coordinators, 2018*) applying significant *E*-value thresholds ($1e^{-5}$). Selected proteins, covering the ZAD over the complete length, were aligned with MAFFT (v7.505, -linsi method) (*Katoh and Toh, 2008*), and the ZAD region extracted with Jalview (*Waterhouse et al., 2009*). A maximum likelihood phylogenetic tree was calculated with IQ-TREE 2 (v.2.2.0) (*Minh et al., 2020*), with standard model selection using ModelFinder (*Kalyaanamoorthy et al., 2017*) and ultrafast bootstrap (UFBoot2) support values (*Hoang et al., 2018*). The tree was visualized in iTOL (v6) (*Letunic and Bork, 2021*). Branches that are supported by an ultrafast bootstrap (UFBoot) value ≥95% are indicated by a gray dot. Branch lengths represent the inferred number of amino acid substitutions per site, and branch labels are composed of gene name (if available), genus, species, and accession number. A similar approach was performed to collect Rhino and HP1 sequences. Full-length *D. melanogaster* HP1-like sequences were used as query for blast searches applying highly significant *E*-value thresholds ($1e^{-10}$). Only sequences covering both chromodomain (CD) and chromo shadow domain (CSD) were considered for further analysis. The alignment was condensed by removing all columns covering less than 70% of the sequences and a maximum likelihood phylogenetic tree was inferred. To search for residues in Rhino that are distinct from all other HP1-like families, we focused on 17 *Drosophila* species where Kipferl could be detected and extracted 104 protein sequences. In the resulting alignment, subfamily specific residues were detected with the multi-relief method (https://www.ibi.vu.nl/programs/shmrwww/; *Brandt et al., 2010*).

## AlphaFold predictions

AlphaFold2-Multimer (*Jumper et al., 2021*; *Evans et al., 2022*) was used to predict protein–protein interactions on a local GPU cluster with a script using MMseqs2 (*Steinegger and Söding, 2017*) (git@92deb92) for local MSA creation and Colabfold (*Mirdita et al., 2022*) (git@7227d4c) for structure prediction. Protein structures were analyzed using ChimeraX (*Pettersen et al., 2021*).

## Expression and purification of the Rhino chromodomain

His$_6$-SUMO-RhinoCD constructs (spanning Rhino residues 20–90 in the vector pET-28) were transformed into the *E. coli* strain BL21-CodonPlus (DE3)-RIPL (Agilent) for large-scale expression using standard methods. Briefly, cultures were grown in Terrific Broth media supplemented with appropriate antibiotic(s) at 37°C to a culture density of approximately OD$_{\lambda=600\ nm}$ of 1.2. Cultures were then cooled in an ice water bath for 15 min followed by induction of protein expression with 0.5 mM Isopropyl ß-D-1-thiogalactopyranoside (IPTG). Induction proceeded overnight at 16°C with shaking at 220 rpm. Cells were harvested by centrifugation at 4000 × *g* for 30 min at 4°C. For Ni-NTA purification, cell pellets were resuspended in 20 ml lysis buffer (50 mM sodium phosphate, pH 8.0, 50 mM NaCl, 10 mM imidazole, 10 µg/ml DNase I, and protease inhibitors) per liter culture. The resuspended cells were lysed by sonication and the lysate was then clarified by ultracentrifugation at roughly 140,000 × g for 30 min. The soluble supernatant was taken for affinity purification via Ni-NTA column (1.5 ml of beads per liter culture), pre-equilibrated with lysis buffer. Beads were washed with 10 column volumes of wash buffer (50 mM sodium phosphate, pH 8.0, 200 mM NaCl, 10 mM imidazole) followed by elution of the target protein in 50 mM sodium phosphate, pH 8.0, 100 mM NaCl, 150 mM imidazole. To remove the affinity tag, Ulp1 protease was added in a 1:10 mass ratio (protease:RhinoCD) and incubated overnight at 4°C. 1 mM Ethylenediaminetetraacetic acid (EDTA) and 5 mM DTT (Dithiothreitol, final concentrations) were added to limit degradation and enhance tag cleavage, respectively. The protein was further purified using tandem ion exchange chromatography with HiTrap Q HP and HiTrap SP HP columns (Cytiva/GE Healthcare Life Sciences). Digested protein was first diluted threefold with low salt buffer (20 mM Tris, pH 7.5, 1 mM DTT) then applied to the HiTrap Q column. The flowthrough was collected and purified using the HiTrap SP column. The target protein was eluted using a 0- to 1-M NaCl gradient in 20 mM Tris, pH 7.5, and 1 mM DTT over approximately 60 ml. Peak fractions were assessed by sodium dodecyl sulfate–polyacrylamide gel electrophoresis (SDS–PAGE) then selected and pooled for further purification. Pooled fractions were concentrated and further purified by gel filtration chromatography using a Superdex75 column equilibrated with 20 mM Tris, pH 7.5, 150 mM NaCl, 1 mM DTT. Depending on the total yield, either a Superdex75increase 10/300 column or a Superdex75 HiLoad 16/600 column (Cytiva/GE Healthcare Life Sciences) was used. Peak fractions were assessed by SDS–PAGE. Fractions with highly purified protein were concentrated, then stored at 4°C. For long-term storage the protein was flash frozen in liquid nitrogen then kept at −80°C. Typical yields were 1–10 mg of purified protein (>98% pure as assessed by SDS–PAGE) per liter culture.

## Size-exclusion chromatography with inline multiangle light scattering

iangle light scattering was used to determine the oligomeric state of the purified proteins. Roughly 400 µg of purified protein (100 µl at 4 mg/ml) was taken for in-line size-exclusion chromatography on a Superdex75increase 10/300 GL column (monitored at 280 nm) followed by light scattering analysis. Chromatography was performed in a buffer of 20 mM Tris, pH 7.5, 150 mM NaCl. MALS was measured with a Wyatt Dawn Heleos-II and processed using the included software (ASTRA Version 5.3.4). Bovine serum albumin (BSA) was used as calibration control.

## Circular dichroism

Circular dichroism was used to assess the folding of the various Rhino chromodomain constructs. Prior to data collection, proteins were exchanged into 10 mM sodium phosphate, pH 7.5, 0.15 M NaF using Zeba 7 kDa spin desalting columns (Thermo Fisher Scientific) then diluted to approximately 50 µM in the desalting buffer. Samples were measured in a 0.2-mm path length demountable quartz cuvette (Hellma) and data were acquired using a Chirascan V100 Spectrometer (Precision Biomolecular Characterization Facility, Columbia University). Spectra were collected at 22°C with a data pitch of 1 nm and scan speed of 1 nm/s. Data shown are the average of three scans after buffer subtraction and presented in units of mean residue ellipticity (degrees·cm$^2$·dmol$^{-1}$·residue$^{-1}$). Fitting was performed by DichroWeb (*Miles et al., 2022*) using the CONTIN-LL method (*Provencher and Glöckner, 1981*) with reference set 3. All fits had an normalized root mean square deviation (NRMSD) of 0.1 or less.

## Isothermal titration calorimetry

Approximately 500 µl of each construct was dialyzed (3.5 kDa molecular weight cutoff) into 20 mM Tris, pH 8.0, 25 mM NaCl, and 2 mM β-Mercaptoethanol overnight at 4°C. The protein concentration

was then determined by absorbance at 280 nm after which the protein was diluted to 100 μM in dialysis buffer. H3K9me3 peptide (KQTAR-K[me3]-STGGK) was purchased from AnaSpec, Inc and resuspended at approximately 1 mM in dialysis buffer. Calorimetry was conducted using a MicroCal iTC200 at 20°C with stirring at 750 rpm with a reference power of 11 μcal/s. Sixteen 2.5 μl injections were performed with an injection spacing of 120 s. Binding curves were analyzed using the included Origin 7 SR4 (version 7.0552 (B552)) software.

## RNA fluorescence in situ hybridization

RNA FISH for *Rsp* and *1.688* g/cm$^3$ Satellites was performed using an in-house labeled probe set composed of 48 oligos or a single fluorescent oligo, respectively (*Wei et al., 2021*; *Gaspar et al., 2017*). RNA FISH for *HMS-Beagle*, *Max*, *diver*, and *3S18* transposons was performed using Stellaris probes (Biosearch Technologies). Probe sequences are listed in *Baumgartner et al., 2022*. Briefly, five pairs of ovaries were dissected into ice-cold phosphate-buffered saline (PBS), fixed at room temperature for 20 min (4% formaldehyde, 0.3% Triton X-100 in PBS), washed three times for 5 min at room temperature (PBS containing 0.3% Triton X-100) followed by incubation at 4°C overnight in 70% EtOH for full permeabilization. Ovaries were rehydrated for 5 min in wash buffer (10% formamide in 2× saline-sodium citrate (SSC) buffer) prior to hybridization, which was done in 50 μl hybridization buffer (100 mg/ml dextran sulfate and 10% formamide in 2× SSC) overnight at 37°C using 0.5 μl *Rsp* FISH probe per sample and a final concentration of 100 nM for the *1.688* g/cm$^3$ FISH oligo. Samples were rinsed twice in wash buffer and washed in wash buffer twice for 30 min at 37°C. Ovaries were counterstained for DNA (DAPI (4',6-diamidino-2-phenylindole) 1:5000 in 2× SSC) for 5 min at RT followed by two washes for 5 min with 2× SSC. Ovaries were mounted on microscopy slides using DAKO mounting medium (Agilent) and equalized at RT for at least 24 hr before imaging on a Zeiss LSM 880 inverted Airyscan microscope. Images are shown as Z-stack across a maximum of 2 μm.

## Immunofluorescence staining of ovaries

Five to ten ovary pairs were dissected into ice cold PBS before fixation (4% formaldehyde, 0.3% Triton X-100, 1× PBS) for 20 min at room temperature with rotation. Fixed ovaries were washed three times for 5 min each in PBX (1× PBS with 0.3% Triton X-100) and blocked with BBX (1× PBS with 0.1% BSA and 0.3% Triton X-100) for 30 min at room temperature with rotation. Incubation with primary antibody was performed at 4°C overnight with antibodies diluted in BBX. After three 5 min washes in PBX, ovaries were incubated overnight at 4°C with fluorophore-coupled secondary antibodies, washed three times in PBX with DAPI in the first wash (1:50,000 dilution). The final wash buffer was carefully removed before addition of DAKO mounting medium. The samples were imaged on a Zeiss LSM 880 confocal-microscope and image processing was done using FIJI/ImageJ (*Schindelin et al., 2012*). Images are shown as Z-stack projection across a maximum of 2 μm. All relevant antibodies and dilutions are listed in the Key Resource Table.

## Scoring of embryo hatching rates

To determine female fertility, 10 females were collected as virgins and aged for 2–3 days with *w$^{1118}$* males. The hatching rate of eggs laid on apple juice plates within 4–7 hr was determined 30 hr after collection (25°C) as the percentage of hatched eggs out of the total. Only plates with more than 50 eggs were included in the analysis. Wildtype females were included as a control.

## Definition and curation of 1-kb genomic windows

Non-overlapping 1-kb tiles were generated based on the four assembled chromosomes of the *D. melanogaster* genome (dm6 assembly) and intersected with genomic piRNA cluster coordinates for annotation. Tiles with a mappability of less than 25%, as determined by intersection with genomic blocks of continuous mappability using BEDTools coverage, were excluded from all analyses (2761 1-kb tiles). In addition, tiles with more than a threefold deviation from the median values for representative input libraries used in *Baumgartner et al., 2022* (18,268 1-kb tiles) or tiles with strong residual Rhino or Kipferl signal in ChIP-seq libraries prepared from the respective knockout ovaries (20 and 495 tiles, respectively) were removed.

## ChIP-seq

ChIP was performed as described previously (*Lee et al., 2006*). In brief, 150 μl of ovaries were dissected into ice-cold PBS, followed by crosslinking with 1.8% formaldehyde in PBS for 10 min at

room temperature, quenching with glycine, and rinsing with PBS. Samples were flash frozen in liquid nitrogen after removing all PBS. Frozen ovaries were disrupted in PBS using a Dounce homogenizer (tight) and centrifuged at low speed. The pellet was resuspended in lysis buffer. Samples were sonicated (Bioruptor) to obtain DNA fragment sizes of 200–800 bp. Samples were incubated with specific antibodies overnight at 4°C in 350–700 µl total volume using 1/4 to 1/3 of chromatin per ChIP (antibodies are listed in Key Resource Table). 40 µl Dynabeads (equal mixture of Protein G and A, Invitrogen) were then added and incubated for 1 hr at 4°C for immunoprecipitation. Following multiple washes, immunoprecipitated protein–DNA complexes were eluted with 1% SDS. Treatment with RNAse-A, decrosslinking overnight at 65°C, and proteinase K treatment were performed before clean-up using ChIP DNA Clean & Concentrator columns (Zymo Research). Barcoded libraries were prepared using the NEBNext Ultra II DNA Library Prep Kit for Illumina (NEB) according to the manufacturer's instructions and sequenced on a NovaSeqSP instrument (Illumina).

## Small RNA-seq

Small RNA cloning was performed as described in *Grentzinger et al., 2020*. In brief, ovaries or testes were lysed and Argonaute-sRNA complexes were isolated using TraPR ion exchange spin columns. sRNAs were subsequently purified using acidic phenol. 3′ adaptors containing six random nucleotides plus a 5 nt barcode on their 5′ end and 5′ adaptors containing four random nucleotides at their 3′ end were subsequently ligated to the small RNAs before reverse transcription, PCR amplification, and sequencing on an Illumina NovaSeqSP instrument.

## Computational analysis

### ChIP-seq analysis

ChIP-seq reads were trimmed to remove the adaptor sequences. Reads were mapped to the dm6 genome using Bowtie (version.1.3.0, settings: -f -v 3 -a `--best --strata --sam`), allowing up to three mismatches. Genome-unique reads were mapped to 1-kb tiles, normalized to library depth, and a pseudocount of '1' was added before enrichment values over input were determined. Each ChIP-seq sample was adjusted using a correction factor based on median input levels and median background levels to reach median background enrichment of 1 to correct for unequal ChIP efficiency. Replicates were averaged for genomic 1-kb tile analyses.

### ChIP-seq analysis on transposon consensus sequences

Genome-mapping reads longer than 23 nucleotides were mapped to TE consensus sequences using bowtie (v.1.3.0; settings: -f -v 3 -a `--best --strata --sam`) allowing up to three mismatches. Reads mapping to multiple elements were assigned to the position with the best mapping. Reads mapping to multiple positions were randomly distributed. To obtain one value per element, library depth-normalized ChIP and input reads were averaged over all nucleotide positions of each element. ChIP-seq enrichment was calculated with a pseudo count of 1 and adjusted using sample-specific correction factors determined from background 1-kb tiles to achieve median background enrichments of 1.

### Small RNA-seq analysis

Raw reads were trimmed for linker sequences, barcodes and the 4/6 random nucleotides before mapping to the *D. melanogaster* genome (dm6), using Bowtie (version.1.3.0, settings: -f -v 3 -a `--best --strata --sam`) with 0 mismatches allowed. Genome-mapping reads were intersected with Flybase genome annotations (r6.40) using BEDTools to allow the removal of reads mapping to rRNA, tRNA, snRNA, snoRNA loci, and the mitochondrial genome. For TE mappings, all genome mappers were used allowing no mismatches. Reads mapping to multiple elements were assigned to the best match. Reads mapping equally well to multiple positions were randomly distributed. Libraries were normalized to 1 million sequenced microRNA reads. For calculation of piRNAs mapping to TEs, only antisense piRNAs were considered, and counts were normalized to TE length. For classification of tiles and transposons into Rhino-independent and -dependent TEs in ovaries and testes, a binary cutoff of at a twofold reduction in antisense piRNA levels in *rhino* knockdown compared to control was applied based on the control samples of the respective tissue.

### Local unique piRNA cluster-mapping piRNAs

piRNA counts of major piRNA clusters relevant in ovaries or testes were determined using cluster definitions established by *Chen and Aravin, 2023*. Locus-unique multi-mappers were obtained by intersecting the 5′ ends of the genome aligned reads with the cluster coordinates. Only reads intersecting only with a single source locus and nowhere else in the genome were allowed. Reads mapping multiple times within one source locus were allowed but only counted once. To account for genotype differences, tiles with a read count of zero in any of the analyzed genotypes were excluded from the analysis.

## Acknowledgements

We thank the NGS, and VDRC units at VBCF, the IMBA/IMP/GMI BioOptics facility and the IMBA Fly House for their invaluable support. We thank Leemor Joshua-Tor for instrument support, Clemens Plaschka for experimental advice, and the Brennecke and Joshua-Tor laboratories for help throughout the project. This research was funded by the Austrian Academy of Sciences, the European Research Council (ERC-2015-CoG-682181 to JB), and the Austrian Science Fund (W1207 to JB). Circular Dichroism spectrophotometry was conducted at the Precision Biomolecular Characterization Facility (PBCF) at Columbia University, supported by NIH award 1S10OD025102-01. LB was funded by a Boehringer Ingelheim Fond PhD Fellowship, JJI was supported by funding from the Howard Hughes Medical Institute, UH was supported through the European Union's Framework Programme for Research and Innovation Horizon 2020 (Marie Curie Skłodowska grant 896416) and through an EMBO long-term fellowship (ALTF_1175-2019).

## Additional information

### Funding

| Funder | Grant reference number | Author |
|---|---|---|
| Boehringer Ingelheim Fonds | | Lisa Baumgartner |
| European Research Council | CoG-682181 | Julius Brennecke |
| Austrian Science Fund | 10.55776/w1207 | Julius Brennecke |
| Howard Hughes Medical Institute | | Jonathan J Ipsaro |
| Horizon 2020 Framework Programme | 896416 | Ulrich Hohmann |
| EMBO long-term fellowship | ALTF_1175-2019 | Julius Brennecke |
| National Institutes of Health | 1S10OD025102-01 | Julius Brennecke |
| European Research Council | ERC-2015-CoG-682181 | Julius Brennecke |

The funders had no role in study design, data collection, and interpretation, or the decision to submit the work for publication.

### Author contributions

Lisa Baumgartner, Conceptualization, Formal analysis, Investigation, Visualization, Methodology, Writing – original draft, Writing – review and editing; Jonathan J Ipsaro, Data curation, Formal analysis, Investigation, Methodology, Writing – review and editing; Ulrich Hohmann, Data curation, Formal analysis, Investigation, Visualization, Writing – review and editing; Dominik Handler, Formal analysis, Visualization, Writing – review and editing; Alexander Schleiffer, Formal analysis, Methodology, Writing – review and editing; Peter Duchek, Resources, Methodology; Julius Brennecke, Conceptualization,

Supervision, Funding acquisition, Investigation, Writing – original draft, Project administration, Writing – review and editing

#### Author ORCIDs
Lisa Baumgartner ![ORCID] https://orcid.org/0000-0001-7769-1274
Jonathan J Ipsaro ![ORCID] https://orcid.org/0000-0002-2743-3776
Ulrich Hohmann ![ORCID] https://orcid.org/0000-0003-2124-1439
Dominik Handler ![ORCID] http://orcid.org/0000-0002-1059-4960
Alexander Schleiffer ![ORCID] http://orcid.org/0000-0001-6251-2747
Julius Brennecke ![ORCID] http://orcid.org/0000-0002-5141-0814

Reviewer #1 (Public Review): https://doi.org/10.7554/eLife.93194.3.sa1
Reviewer #3 (Public Review): https://doi.org/10.7554/eLife.93194.3.sa2
Author response https://doi.org/10.7554/eLife.93194.3.sa3

---

## Additional files

### Supplementary files
- Supplementary file 1. Protein accessions used for Rhino chromodomain alignments.
- Supplementary file 2. Previously published ChIP data sets analyzed in this study.
- MDAR checklist

### Data availability
Sequencing datasets have been deposited to the NCBI GEO archive (GSE244196). Previously published data sets analyzed in this study are listed in Supplementary File 2. All fly strains generated for this study are available via the VDRC (http://stockcenter.vdrc.at/control/main).

The following dataset was generated:

| Author(s) | Year | Dataset title | Dataset URL | Database and Identifier |
|---|---|---|---|---|
| Baumgartner L, Ipsaro JJ, Hohmann U, Handler D, Schleiffer A, Duchek P, Brennecke J | 2023 | Evolutionary adaptation of the chromodomain of the HP1 protein Rhino allows the integration of heterochromatin and DNA sequence signals | https://www.ncbi. nlm.nih.gov/geo/ query/acc.cgi?acc= GSE244196 | NCBI Gene Expression Omnibus, GSE244196 |

The following previously published dataset was used:

| Author(s) | Year | Dataset title | Dataset URL | Database and Identifier |
|---|---|---|---|---|
| Baumgartner L, Handler D, Platzer SW, Yu C, Duchek P, Brennecke J | 2022 | The *Drosophila* ZAD zinc finger protein Kipferl guides Rhino to piRNA clusters | https://www.ncbi. nlm.nih.gov/geo/ query/acc.cgi?acc= GSE202468 | NCBI Gene Expression Omnibus, GSE202468 |

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
