## [Editor Report · eLife assessment]

This **fundamental** work has completed our understanding of the singular binding profile of the Rhino HP1 protein to chromatin, a key step in converting certain genomic regions into piRNA source loci. The evidence supporting the conclusions is **compelling**. Phylogenetic analyses, structure prediction, rigorous biochemical assays and in vivo genetics emphasize the importance of the Rhino chromodomain in the recognition of both a histone mark and a DNA-binding protein, and highlight the importance of a single chromodomain residue in the protein-protein interaction.

---

## [Referee Report · Reviewer #1 (Public Review)]

Summary:

The manuscript focuses on an unexpected finding that a tiny change in a protein's aminoacid sequence can redefine its biological function. The authors' data and analyses explain how a chromodomain, typically implicated in interactions with histones, can also mediate binding of HP1 homolog Rhino to the non-histone partner protein Kipferl. They elegantly pinpoint the capacity for such interaction to a single aminoacid substitution (in fact, a single-nucleotide! substitution).

Strengths:

Both genetic and biochemical approaches are applied to rigorously probe the proposed explanation. The authors find their predictions to be borne out both in vivo, in mutant animals, and in biochemical experiments. The manuscript also features phylogenetic comparisons that put the finding into a broader evolutionary perspective.

Weaknesses pointed out in the original submission were addressed in the revised manuscript.

---

## [Referee Report · Reviewer #3 (Public Review)]

Summary:

This article is a direct follow-up to the paper published last year in eLife by the same group. In the previous article, the authors discovered a zinc finger protein, Kipferl, capable of guiding the HP1 protein Rhino towards certain genomic regions enriched in GRGGN motifs and packaged in heterochromatin marked by H3K9me3. Unlike other HP1 proteins, Rhino recruitment activates the transcription of heterochromatic regions, which are then converted into piRNA source loci. The molecular mechanism by which Kipferl interacts specifically with Rhino (via its chromodomain) and not with other HP1 proteins remained enigmatic.

In this latest article, the authors go a step further by elucidating the molecular mechanisms important for the specific interaction of Rhino and not other HP1 proteins with Kipferl. A phylogenetic study carried out between the HP1 proteins of 5 *Drosophila* species led them to study the importance of an AA Glycine at position 31 located in the Rhino chromodomain, an AA different from the AA (aspartic acid) found at the same position in the other HP1 proteins. The authors then demonstrate, through a series of structure predictions, biochemical and genetic experiments, that this specific AA in the Rhino-specific chromodomain explains the difference in the chromatin binding pattern between Rhino and the other *Drosophila* HP1 proteins. Importantly, the G31D conversion of the Rhino protein prevents interaction between Rhino and Kipferl, phenocopying a Kipfer mutant.

Strengths:

The strength of this study is to test at the molecular and genetic level whether the difference in the AA sequence- encovered by phylogenetic analysis of HP1 proteins including Rhino combined with structure prediction- can explain the difference in chromatin binding patterns between HP1 proteins and Rhino.

To do so they have created a Rhino mutant by introducing a point mutation into the endogenous rhino gene, reverting the Glycine in position 31 to the aspartic acid found in all other HP1 proteins. Even if the Rhino G31D mutant retains its ability to interact with H3K9me3 (predictive and biochemistry approaches that I'm less familiar with) it does not localize correctly on the chromatin preventing certain regions such as locus 80F from being converted into piRNA source loci. However other regions such as satellite regions attract the Rhino mutant protein converting them into super piRNA source loci, phenocopying the effects observed in a Kipferl mutant. Why Rhino when not bound to Kipferl concentrates in satellite regions is a question that remains unanswered.

Weaknesses:

In this new version of the manuscript, the authors have answered all the questions and weaknesses raised previously.

---

## [Author Response]

The following is the authors’ response to the original reviews.

**Public Review:**
This article is a direct follow-up to the paper published last year in eLife by the same group. In the previous article, the authors discovered a zinc finger protein, Kipferl, capable of guiding the HP1 protein Rhino towards certain genomic regions enriched in GRGGN motifs and packaged in heterochromatin marked by H3K9me3. Unlike other HP1 proteins, Rhino recruitment activates the transcription of heterochromatic regions, which are then converted into piRNA source loci. The molecular mechanism by which Kipferl interacts specifically with Rhino (via its chromodomain) and not with other HP1 proteins remained enigmatic.In this latest article, the authors go a step further by elucidating the molecular mechanisms important for the specific interaction of Rhino and not other HP1 proteins with Kipferl. A phylogenetic study carried out between the HP1 proteins of 5 *Drosophila* species led them to study the importance of an AA Glycine at position 31 located in the Rhino chromodomain, an AA different from the AA (aspartic acid) found at the same position in the other HP1 proteins. The authors then demonstrate, through a series of structure predictions, biochemical, and genetic experiments, that this specific AA in the Rhino-specific chromodomain explains the difference in the chromatin binding pattern between Rhino and the other *Drosophila* HP1 proteins. Importantly, the G31D conversion of the Rhino protein prevents interaction between Rhino and Kipferl, phenocopying a Kipferl mutant.Strengths:The authors' effective use of phylogenetic analyses and protein structure predictions to identify a substitution in the chromodomain that allows Rhino's specific interaction with Kipferl is very elegant. Both genetic and biochemical approaches are applied to rigorously probe the proposed explanation. They used a point mutation in the endogenous locus that replaces the Rhino-specific residue with the aspartic acid residue present in all other HP1 family members. This novel allele largely phenocopies the defects in hatch rate, chromatin organization, and piRNA production associated with kipferl mutants, and does not support Kipferl localization to clusters. The data are of high quality, the presentation is clear and concise, and the conclusions are generally well-supported.Weaknesses:The reviewers identified potential ways to further strengthen the manuscript.(1) The one significant omission is RNAseq on the rhino point mutant, which would allow direct comparison to cluster, transposon, and repeat expression in kipferl mutants.

In this eLife Advances submission, we aim to elucidate the molecular interaction between Rhino and the zinc finger protein Kipferl and how it evolved. Using various assays, of which piRNA sequencing is the most relevant and comprehensive, we show that the rhino[G31D] mutation phenocopies a rhino loss-of-function situation for Kipferl and a kipferl loss-of-function situation for Rhino. Further confirmation of this statement by additional RNA-seq experiments to probe the extent of selective TE de-repression would indeed be a possibility. We decided to test for TE de-repression phenotypes using sensitive RNA-FISH experiments of a handful of TEs that are deregulated in kipferl loss of function flies (Baumgartner at al. 2022). This showed that the same TEs are also deregulated in rhino[G31D] flies, further confirming the similarity of the two genotypes. We have added these data to the text and to Figure 5-figure supplement 2, which shows representative RNA FISH images.

(2) The manuscript would benefit from adding more evolutionary comparisons. The following or similar analyses would help put the finding into a broader evolutionary perspective:i) Is Kipferl's surface interacting with Rhino also conserved in Kipferl orthologs? In other words, are the Rhino-interacting amino acids of Kipferl under any pressure to be conserved?

We performed an analysis of the Kipferl interface that interacts with the Rhino chromodomain in those species where Kipferl could be unambiguously identified. This showed that the residues involved in the Rhino interaction are generally conserved. We have added this analysis to Figure 1-figure supplement 4.

ii) The remarkable conservation of Rhino's G31 is at odds with the arms race that is proposed to be happening between the fly's piRNA pathway proteins and transposons. Does this mean that Rhino's chromodomain is "untouchable" for such positive selection?

We agree that the conservation of the G31 residue argues against this binding interface being under positive selection in Rhino. Without understanding the pressures acting on Rhino that underlie the previously published positive selection, we find it difficult to draw firm conclusions. Mutating G31 in fly species that lack Kipferl would be an interesting experiment.

**Recommendations for the authors:**
(1) RNAseq is important to the full characterization of the phenotype and should be included. It's now clear that the major piRNA clusters are not required for fertility, so I would also include an analysis of piRNA production and Rhino binding to regions flanking isolated insertions.

See our response to raised weakness #1 above. Briefly, we have now added an analysis of TE de-repression based on RNA-FISH experiments (Figure 5-figure supplement 2). Regarding the proposed analysis of piRNA production and Rhino binding to regions flanking isolated TE insertions: this is an important issue that we carefully analysed in our previous work characterising the kipferl mutant (Baumgartner et al. 2022). In the present work, we focused on generating a rhino mutant that uncouples Rhino from Kipferl.

(2) The authors do not provide direct biochemical evidence that the chromodomain substitution blocks Rhino binding to Kipferl. However, Rhino protein is very low abundance, making analysis of the endogenous protein very difficult.

Based on our previous work (Baumgartner et al 2022), the Rhino chromodomain interacts directly with the fourth zinc finger of Kipferl. Mutation of a single residue in the predicted interface (Rhino[G31D]) phenocopies a kipferl mutant, strongly suggesting that this mutation disrupts the Rhino-Kipferl interaction. Definitive evidence will have to await the reconstitution of this interaction using recombinant proteins. Our attempts to purify recombinant Kipferl (expressed in bacteria or in insect cells) or the protein fragments relevant to the interaction were unsuccessful so far. While we obtained soluble fractions of the first ZnF array, there was always a high level of co-purifying nucleic acids that we were not able to remove.

(3) Even if the Rhino G31D mutant retains its ability to interact with H3K9me3 it does not localize correctly on the chromatin preventing certain regions such as locus 80F from being converted into piRNA source loci. However other regions such as satellite regions attract the Rhino mutant protein converting them into super piRNA source loci, phenocopying the effects observed in a Kipferl mutant. Why Rhino when not bound to Kipferl concentrates in satellite regions is a question that remains unanswered.

This is a very interesting question indeed. We have not been able to elucidate the molecular basis of how Rhino is recruited to satellite repeats in Kipferl mutants. For example, we performed a proximity biotinylation experiment with GFP-Rhino in Kipferl mutant ovaries, but this experiment did not reveal any protein that would explain the observed accumulation of Rhino at the complex satellite repeats.

(4) In the phylogenetic analysis the authors identified two residues as Rhino-specific and conserved sequence alterations, the D31G mutation and the G62 insertion. However, the authors limit their study to D31G mutation, and nothing is performed on the G32 insertion. It would have been interesting to know the impact of this insertion on Rhino's biology.

The role, if any, of the Rhino-specific G62 insertion and its effect on Rhino localisation or function is an interesting topic for further study. We have not investigated the G62 residue experimentally. In the current manuscript, we limited our efforts to the analysis of the G31D mutation, as the goal was to identify the mode of interaction with Kipferl, and the G62 residue is not predicted to contact Kipferl according to AlphaFold.

(5) The authors report that the G31D mutation of Rhino phenocopies the Kipferl mutant. Rhino is wrongly localized in the nucleus, and Rhino G31D recruitment in certain Kipferl-enriched regions is affected, as at the 80F locus, which correlates with a strong drop in piRNA production from this locus. To go a step further in demonstrating that G31D phenocopies the Kipferl mutant, it would have been informative to analyse how much TE piRNAs are affected and whether TEs are deregulated.

See our response to similar comments above. We have added RNA-FISH experiments to illustrate that the TE de-repression phenotypes are comparable between rhino[G31D] and kipferl loss of function ovaries (Figure 5-figure supplement 2). Analyses of TE-mapping piRNAs also show well correlated phenotypes (Figure 5-figure supplement 1).

(6) Figure 3A: To homogenize with the immunostaining presented in Figure 3B, can the authors add on the bar graph depicting female fertility the results obtained with kipferl-/- and rhino-/- genotype?

*rhino* mutants are completely (100%) sterile and the fertility of *kipferl* mutants was previously measured to range between 15% and 40% (Baumgartner et al. 2022).

(7) Figure 4A: It would have been interesting to show Venn diagrams showing the overlap of genomic regions enriched for Kipferl versus regions enriched for Rhi in a WT and in a Rhi G31D mutant.

We consider the analysis presented in Figure 4 to be more meaningful, as a Venn diagram would require binary cut-offs.

(8) Figure 1B: In the phylogenic analysis for Rhino/HP1d two D. simulans lines are presented. Can the authors clarify this point?

There are two Rhino paralogs in D. simulans: one paralog (NCBI: AAY34025.1) is more similar to *D. melanogaster* Rhino, contains one intron and is located at chromosome chr2R (assembly Apr. 2005, WUGSC mosaic 1.0/droSim1: 12256895-12258668). The second paralog (XP_002106478.1) is located on chromosome X (6734493-6735248) and does not contain an intron. We have added a clarifying statement to the corresponding figure legend.

(9) To determine whether Rhino G31D point mutation affects the overall function of Rhino, the authors analysed Kipferl-independent piRNA source loci by looking at Responder and 1,688 family satellites. I'm not sure that these loci can be classified as Kipferl-independent piRNA source loci since a strong increase of piRNA production from these loci in Kipferl mutant is observed. In my point of view, the 42AB and 38C are real Kipferl-independent piRNA source loci as piRNA production from these loci is not affected by Kipferl KD.

Indeed, the *Rsp* and *1,688* family satellites are not completely independent of Kipferl, as their expression and Rhino occupancy differ between wild-type and *kipferl* loss-of-function phenotypes (including *rhino[G31D]*). However, we believe that this increase is due to a strong dependence on different sequestration mechanisms and is not mediated by a direct function of Kipferl at these sites. Similarly, we observe slight differences in piRNA production for the peripheral parts of cluster 42AB, as well as differences in Rhino occupancy despite an unaltered piRNA profile at cluster 38C (Baumgartner et al. 2022). Thus, different flavours of Kipferl-independence exist, with the only truly Kipferl-independent piRNA sources likely to be the piRNA clusters in the testis. A clear classification is further complicated by previously observed compensatory effects in the piRNA pathway, leading us to adopt the current definition of "requiring Kipferl for Rhino recruitment" to distinguish Kipferl-dependent from Kipferl-independent sites.

(10) The authors report that the G31D mutation of Rhino phenocopies the Kipferl mutant. Rhino is wrongly localized in the nucleus, and Rhino G31D recruitment in certain Kipferl-enriched regions is affected, as at 80F locus, which correlates with a strong drop in piRNA production from this locus. To go a step further in demonstrating that G31D phenocopies the Kipferl mutant, it would have been interesting to look at how much TE piRNAs are affected and whether TEs (and which class of TE) are deregulated by RNAseq and/or in situ hybridization.

See our response to similar comments above. Our new RNA-FISH experiments and TE-mapping piRNA analysis extend the comparison of phenotypes between kipferl mutants and rhino[G31D] mutants and are consistent with our previous conclusions (Figure 5-figure supplements 1 and 2).